

# *$^{14}$C plateau tuning – A misleading approach or trendsetting tool for*
# *marine paleoclimate studies?*
*Michael Sarnthein[1] and Pieter M. Grootes[2]*
1) Institute of Geosciences, University of Kiel, Olshausenstr. 40, 24098 Kiel, Germany,
michael.sarnthein@ifg.uni-kiel.de
2) Institute of Ecosystem Research, University of Kiel, Olshausenstr. 40, 24098 Kiel,
Germany, pgrootes@ecology.uni-kiel.de
*Corresponding author*:
Michael Sarnthein, michael.sarnthein@ifg.uni-kiel.de,
ABSTRACT
On the basis of minor time scale adjustments including the synchronization on IntCal20 the
Suigetsu-based atmospheric $^{14}$C plateau structures are shown to be authentic. Their global
significance is demonstrated by the coherence with the tree ring record 10 to 15 cal. ka and by
coherent features in the $^{14}$C record of Hulu Cave back to 35 cal. ka. The suite of atmospheric
structures can be recognized in high-resolution ocean sediment records independent of various
processes leading to partial distortion of a sediment record. This provides a unique tool for
global stratigraphic correlation and paleoceanographic studies as shown by supplementary
figures and tables from 19 cores obtained from key locations in the world ocean.




## INTRODUCTION


Sarnthein et al. (2020) gave a synthesis of the growing evidence of the value of [14]C plateau
tuning (PT) for the chronostratigraphic correlation of last glacial-to-deglacial paleoclimate events
in marine proxy records to each other, to climate events recorded in ice cores, and to
speleothems, moreover, to well-dated terrestrial climate records. Bard and Heaton (2021) (B&H)
published a follow-up paper that fundamentally denunciates concept and techniques of PT. The
concerns of B&H were based on perceived problems with the atmospheric plateau structures we
observe for the (Suigetsu-based) radiocarbon record and came from various paleoclimatic,
paleoceanographic, sedimentological, and statistical perspectives. Their critique provided a rare
opportunity to discuss and clarify the (perceived) flaws and weaknesses and strengths of PT. In
CP Discussions (Grootes and Sarnthein, 2021; Sarnthein and Grootes, 2021) we rejected the
critique as it was largely based on:
1) A fundamental misunderstanding of plateau identification: This is not based on the
identification of any single plateau but rather on the best pattern match of a complete suite of
plateaus; much like correlating ice or sediment core-based $\delta^{18}O$ records.
2) B&H's focus on the physics of surface ocean [14]C fluctuations, using a 1998 box-model, instead
of on those of the [14]C difference between atmosphere and surface ocean (MRA).
The B&H response in CP Discussions and their paper, largely ignore this reasoning as well as our
detailed answers to their 17 points of critique, stick to an incorrect perception of the basic
assumptions used for PT and avoid a real discussion.

We here present evidence for the authenticity of Suigetsu [14]C plateaus as atmospheric [14]C
structures that can be observed worldwide. Also, we show that analogous [14]C structures in
ocean planktic sediment records are not merely a result of various sedimentation processes such

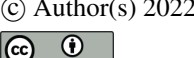



as bioturbational mixing. Finally, to foster future use and discussion of PT, we add as
supplementary material a comprehensive set of tables and figures for all 19 ocean sediment
records published so far on the new IntCal20 time scale.

***Atmospheric $^{14}$C plateaus and jumps are reproduced by diverse statistical approaches***
We here present our Suigetsu-based plateau and jump structures, plotted in the $\Delta^{14}$C domain,
together with the atmospheric $\Delta^{14}$C wiggles 14–30 cal. ka as defined by a Bayesian spline over
the total of Suigetsu $^{14}$C ages (shown with 1σ by B&H 2021, Fig. 3a), now all plotted vs. IntCal20
ages (Bronk Ramsey et al., 2020) (Fig. 1 and Table 1). Moreover, after a careful check of the
position of plateau boundaries, we slightly revised three of them, (i) a slight upward shift of the
base of plateau 7, (ii) omission of the boundary between plateaus 6b and 6a, and (iii) a minimal
backward shift of the 5a-b plateau boundary by a single age date. To our surprise and
satisfaction, the unification of time scales used plus these minimal changes resulted in a basic
upgrade and substantial overlap of the atmospheric $^{14}$C structures revealed by the two
independent methods, the Bayesian spline and the visual inspection of $^{14}$C plateaus and jumps
back to 27 cal. ka. This agreement on authentic structures of atmospheric $^{14}$C may be regarded
as a corner stone crucial to justify PT as legitimate tool for stratigraphic correlation.

Prior to 27 cal ka, however, the raw Suigetsu $^{14}$C ages for plateaus 10b and 11, generated by
different laboratories, are diverging by up to 1000 yr at analytical 1σ uncertainties of ~150 to
<400 years (Bronk Ramsey et al., 2012), a discrepancy that our plateau definitions tried to bridge
by weighted average values, the Bayesian spline, however, has valued differently.


The plateau structures of the Suigetsu atmospheric [14]C record are clearly paired with IntCal20
tree ring- and floating tree ring-based [14]C structures for the interval 10 - ~15 cal. ka (Fig. 2; suppl.
by Adolphi et al., 2017). Thus Fig. 2 can positively answer the question, raised by B&H (2021),
whether the [14]C plateau structures defined by Sarnthein et al. (2020) in the Lake Suigetsu record
present a suite of authentic features of atmospheric [14]C, that indeed can be globally reproduced.
For comparison, the smoothed character of the IntCal20 curve beyond 14 ka is due to a change
in the available calibration data rather than to a fundamental change in the atmosphere (Reimer
et al., 2020; B&H, 2021). This is supported by a comparison of Bayesian spline compilations of
the Suigetsu (green) and the Hulu speleothem (magenta) datasets over the period 20-30 cal. ka,
with Hulu deconvoluted using a MatLab algorithm (Fig. 3 from Bronk Ramsey et al., 2020).

Consequentially, we regard it legitimate to extrapolate our interpretation of fine structures in
the Suigetsu [14]C record further back, at least up to 27 cal. ka (Fig. 4). That is, we may assume that
prior to 15 cal. ka the continuing [14]C fine structures of the admittedly somewhat noisy Suigetsu
record of atmospheric [14]C jumps and plateaus come close to reality per analogy with the match
with the tree ring record 10 to ~15 cal. ka. Our reasoning is supported independently by various
Suigetsu plateaus between 15 and 30 cal. ka and structures further back (e.g., parts of plateaus
number 2b, 4, 6, 8, 9; Fig. 4) that are largely reproduced also by a Bayesian spline of analogous
[14]C plateau structures in the deconvoluted U/Th-dated [14]C record based on Hulu Cave data (Fig.
3; from Bronk Ramsey et al., 2020). Altogether, the Suigetsu record of atmospheric [14]C wiggles
provides over its full length a suitable target for global correlation. This is more authentic than
the better defined [14]C trend of the Hulu speleothem and the IntCal20 records that, admittedly,
have been smoothed.



Objections against the use of this Suigetsu [14]C record instead of IntCal20 [14]C ages as basis for the
definition of atmospheric [14]C plateaus and jumps, as formulated by B&H 2021, ignore a crucial
difference between PT and [14]C calibration. *Calibration* aims to provide the best possible estimate
for the 'calendar' age corresponding to a [14]C age. This estimate is provided by IntCal20, the
collection of pointwise averages of 2500 Bayesian spline realizations of the [14]C calibration curve,
based on all sorts of available [14]C ages prior to 14 cal. ka. Statistically integrating Suigetsu and
floating tree ring ages, as purely atmospheric record, with a number of carbonate-based coral-,
marine sediment-, and speleothem-based records, results in a statistically secure but smoothed
IntCal20 record (B&H Fig. 4b).

*PT* is a research tool that employs a suite of medium well age-calibrated structures in the
Suigetsu atmospheric [14]C record as global reference to explore the [14]C fine structure of noisy,
local ocean plankton records (Suppl. Materials Fig. S1-S19). This is a new but trendsetting
approximation to obtain a new order of age tie points for centennial-to-millennial-scale global
stratigraphic correlation and a major addition to the role of radiocarbon as key tracer (Heaton et
al., 2021).

Both on the basis of visual inspection and the 1st derivative of a [14]C record (see Figs. S1–S19) the
robustness and uncertainty of the age tie points are best calibrated at the marked [14]C-age jumps
that separate two subsequent [14]C plateaus each, the range of which is marked by enveloping
'boxes' in Fig. 4. Accordingly, the uncertainty in the cal. age of the beginning and end of a [14]C
jump/plateau hardly exceeds ±50 to ±100 years, when employing the age estimates listed by
Bronk Ramsey et al. (2020).



Uncertainty levels of the exploratory PT chronology are, inevitably, somewhat higher than those
acceptable for a $^{14}$C calibration. This concerns the identification of plateaus as well as the
definition of plateau boundaries in some less densely sampled planktic $^{14}$C records (c.f. Figs. S1-
S19). In view of the identification Sarnthein et al. (2020) again emphasized that the validity of
certain atmospheric $^{14}$C plateaus assigned to structures in a single sediment $^{14}$C record must be
verified by detailed comparison with coarser-spaced 'conventional' stratigraphic tie points in an
ocean sediment record such as those provided by planktic $\delta^{18}$O records (e.g., DO event 1),
turning points in sea surface temperature, tephra layers (e.g., Fig. S16).

***Correlation of atmospheric and planktic $^{14}$C records***
With the reproducibility of Suigetsu atmospheric $^{14}$C concentration patterns and their value as a
tool for PT global correlation studies established, the question remains whether high-resolution
planktic sediment $^{14}$C records indeed reflect primarily the atmospheric $^{14}$C structures defined in
the Suigetsu record. B&H (2021) explored this question in their modeling of hypothetical
'Suigetsu' and 'Cariaco' records derived by adding appropriate noise to a section of the IntCal20
tree ring record. Their calculations indicated that no statistically robust signal could be extracted
which, as explained above, does not preclude the use of PT as an exploratory tool for age
correlation. B&H failed to consider the full period 10-~15 cal ka where tree ring-based $^{14}$C
structures overlap with Suigetsu data and restricted their tree-ring comparison to the less
informative section 12.0-13.9 ka. Yet, even for this section their modeled 'Cariaco' curves
indicate the underlying IntCal20 tree-ring $^{14}$C fluctuations that could be used by PT to explore
the age correlation of such a record.



Two important questions raised by B&H (2021) were (i) Can $^{14}$C structures observed in planktic
$^{14}$C records result 'accidentally' from various processes characteristic of sediment deposition and
bioturbation? and (ii) Can a reliable correlation between atmospheric and planktic $^{14}$C plateaus
be made?

B&H's repeated objection to plateau identification in marine sediment cores was based on
disturbance by bioturbational mixing. This objection is invalid for three reasons:
-- At high latitudes and water depths >3000 m (with reduced flux of Corg) the bioturbational
homogenic mixing depth amounts to 2-3 cm (Trauth et al., 1997). This won't affect $^{14}$C signals at
average sedimentation rates of 10 - 50 cm/ky.  At these high rates mixing depths even reaching
7-10 cm in low latitudes are little relevant, also shown by paired trends of quasi-continuous, e.g.,
XRF-based proxy records.
-- Wide sediment sections in five out of 19 cores in Sarnthein et al. (2020; cores MD3180,
MD2503, ODP1002D, ODP893A, PS97-137; Suppl. Figs. S3, S4, S11, S10, S15; Table S20) are
laminated, thus largely free of bioturbational mixing.
-- Planktic species counts in two high-resolution, non-laminated cores (GIK23074, SHAK06-5K)
refute any age offsets of the $^{14}$C plateau signal due to differential bioturbation.

Results of the earlier use of PT on planktic sediment $^{14}$C records, now converted to the Bronk-
Ramsey et al, (2020) time scale, are presented for comparison as supplemental material (Suppl.
Materials Fig. and Tables S1-S19). Though sediment deposition and bioturbational processes
occasionally locally affect the sediment $^{14}$C records, the full pattern of sequential $^{14}$C fluctuations
generally allows a reliable correlation.



The age tie points provided by [14]C PT bring more age control to the sediment records and, in
doing so, reveal fluctuations in sediment deposition that were hitherto undefined as well as
coeval variations in marine reservoir ages (MRA). Such short-term and local small-scale changes
in sedimentation rate and MRA critically depend on the potential plateau numbers assigned to a
plateau suite by alternative models of PT. The choice of tuning model, finally, is carefully based
on and constrained by conventional sediment properties and stratigraphic tie points (e.g., Fig.
S16a). In case of persisting alternative age models, we prefer the model where sedimentation
rates and MRA's show the lowest fluctuations over a suite of subsequent plateaus (e.g., Fig. S5)
(Sarnthein et al., 2007).

Most of the gaps and lows in sediment deposition defined by PT are already indicated by major
age jumps that mark the record of raw [14]C ages, independent of any PT (Suppl. Materials, Figs.
S2, S6, S10, S13 to S19). Short-term major changes of sedimentation rate have been established
independently by 230Th-based high-resolution age control for North Atlantic sediment cores
(Missiaen et al. 2019), which supports the reality of the sediment fluctuations revealed by PT.
Most sedimentation spikes, moreover, make sense in terms of paleoceanography and
paleoclimate. Pertinent changes in sedimentation rate may thus be derived by PT, but may also
serve as corrective to obtain a best possible tuning of atmospheric and marine sediment-based
plateau suites.

CONCLUSION
On the basis of coherence with tree ring records 10 to 15 cal. ka we can conclude that the
pattern of [14]C fluctuations in the atmospheric Suigetsu record represents an atmospheric [14]C
signal that can be used for global correlation with a precision better than ±50 to ±100 years. The



PT technique explores the detailed dating of planktic $^{14}$C records by correlation to the Suigetsu
signal, requiring it to be consistent with the available conventional evidence of stratigraphic
correlation. The results of PT provide new insights into local marine reservoir ages (MRA), thus
to local oceanography by revealing short-term changes in sedimentation rate and regional ocean
mixing. These are important tracers for studies of paleoceanography and paleoclimatic events
on a global atmospheric time scale. Such changes are often missed by the widely employed
wide-spaced set of conventional age tie points and related average sedimentation of ocean
sediment records.

**ACKNOWLEDGMENTS**
We are grateful to S. Beil, Kiel, for generous computer assistance.

**Data availability**

All primary radiocarbon data and cal. ages assigned are stored at PANGAEA.de® under , , , ,

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



**TABLES and FIGURES**
Table 1 a and b. Summary of recently revised U/Th model-based age estimates (Bronk Ramsey et
al., 2020) for ~30 plateau (pl.) boundaries in the atmospheric [14]C record identified in Lake
Suigetsu Core SG06 by means of visual inspection over the interval 10.5–27 cal. ka (Sarnthein et
al., 2015, suppl. and modified). At the right-hand side, three columns give the average (Ø) and
uncertainty range of [14]C ages for each [14]C plateau.

| SUIGETSU SG06_2012 Plateau no. | Plateau Top IntCal20 U/Th-based age (yr BP) | Depth (cm c.d.) | Plateau Base IntCal20 U/Th-based age (yr BP) | Depth (cm c.d.) | Ø 14C Age of 14C Plateau (14C yr) | ±Uncertainty (14C yr) | 14C age BP min/max. (1.6 σ range) |
|---|---|---|---|---|---|---|---|
| 'Preboreal' | 10560 | 1325 | 11108 | 1383 | 9525 | −170/+110 | 9356/ 9635 |
| 'Top YD' | 11281 | 1402 | 11755 | 1453 | 10060 | −100/+35 | 9963/ 10095 |
| 'YD' | 11895 | 1467 | 12475 | 1525 | 10380 | −170/ 124 | 10211/ 10504 |
| 'no name' | 12780 | 1555 | 13080 | 1582 | 11000 | −85/ 114 | 10915/ 11114 |
| 1a | 13656 | 1626 | 14065 | 1657 | 12006 | 100 | 11857/ 12050 |
| 1 | 14187 | 1666 | 15044 | 1740 | 12471 | 185 | 12315/ 12683 |
| 2a | 15415 | 1754 | 16531 | 1802 | 13406 | 245 | 13174/ 13665 |
| 2b | 16531 | 1802 | 16940 | 1820 | 13850 | 40 | 13808/ 13885 |
| 3 | 17579 | 1847 | 18189 | 1888 | 14671 | 105 | 14582/ 14792 |
| 4 | 18790 | 1913 | 19793 | 1971 | 15851 | 190 | 15661/ 16044 |


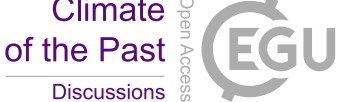

| | | | | | | | |
|---|---|---|---|---|---|---|---|
| 5a | *19922* | 1978 | *20240* | 2003 | **16670** | 90 | 16570/16750 |
| 5b | *20240* | 2003 | *20919* | 2032 | **17007** | 190 | 16830/17247 |
| 6 | *21173* | 2105 | *22300* | 2132 | **17766** | 404 | 17433/18240 |
| 7 | *22604* | 2140 | *22940* | 2171 | **18844** | 117 | 18741/18975 |
| 8 | *23237* | 2175 | *24300* | 2257 | **19715** | −290 325 | 19425/20041 |
| 9 | *24300* | 2257 | *25250* | 2312 | **20465** | −227 263 | 20238/20728 |
| 10a | *25656* | 2358 | *26960* | 2400 | **22328** | −380 270 | 21946/22600 |
| 10b | *26960* | 2400 | *27612* | 2426 | **22708** | −475 440 | 22233/23147 |
| 11 | *27900* | 2443 | *28898* | 2525 | **24088** | −360 505 | 23727/24595 |





Figure 1. Raw ¹⁴C data of Lake Suigetsu (blue dots), converted into ‰ Δ¹⁴C units, plotted vs. cal.
ages of Bronk Ramsey et al. (2020). A Bayesian spline named "Suigetsu only curve" (pink band,
B&H 2021, Fig. 3a modified) shows periods of gradually decreasing atmospheric Δ¹⁴C values,
reflecting 15 atmospheric ¹⁴C age plateaus and their uncertainty range as defined by Sarnthein
et al. (2020; numbers listed in Table 1). In between, rapidly increasing atmospheric Δ¹⁴C values
reflect short gaps or ¹⁴C age 'jumps' between plateaus. Superimposed are straight lines that
display the atmospheric Δ¹⁴C structures as originally defined by visual inspection of raw ¹⁴C ages
and 1st derivative technique (Sarnthein et al., 2020; and Suppl. Figs. 1-19), with green lines using
the initial chronology of Bronk Ramsey et al. (2012; B&H 2021) and dark pink lines using the
recently revised ages of Bronk Ramsey et al. (2020).

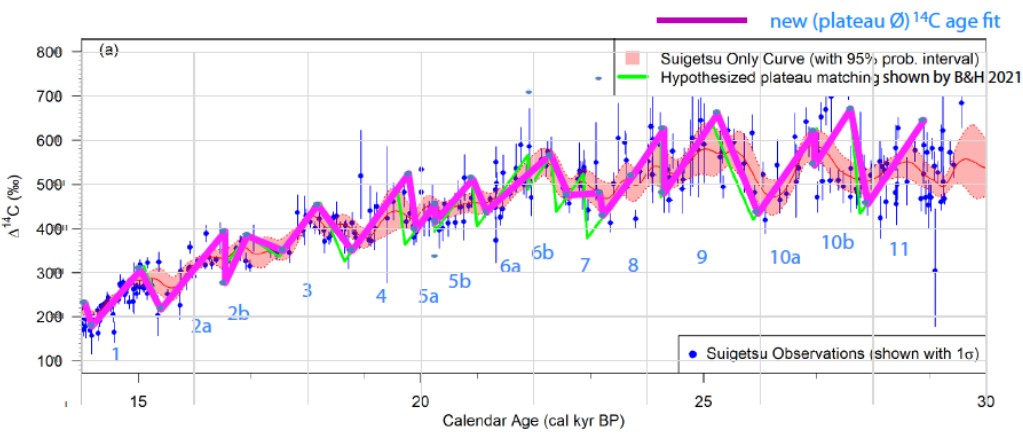



Figure 2. High-resolution record of atmospheric ¹⁴C jumps and plateaus (= suite of labeled
horizontal boxes that envelop scatter bands of largely constant ¹⁴C ages extending over
>300 cal. yr) in a sediment section of Lake Suigetsu (Fig. 2 of Sarnthein et al., 2020) vs.
tree ring-based ¹⁴C jumps and plateaus 10–14.5 cal. ka (Reimer et al., 2013; 14.0-14.4
cal. ka: suppl. by data of Adolphi et al., 2017). Blue line averages paired double and



triple [14]C ages of Suigetsu plant macrofossils. Age control points (cal. ka) follow varve
counts (Schlolaut et al., 2018) and U/Th model-based ages of Bronk Ramsey et al. (2012
and 2020). YD = Younger Dryas, B/A = Bølling-Allerød.

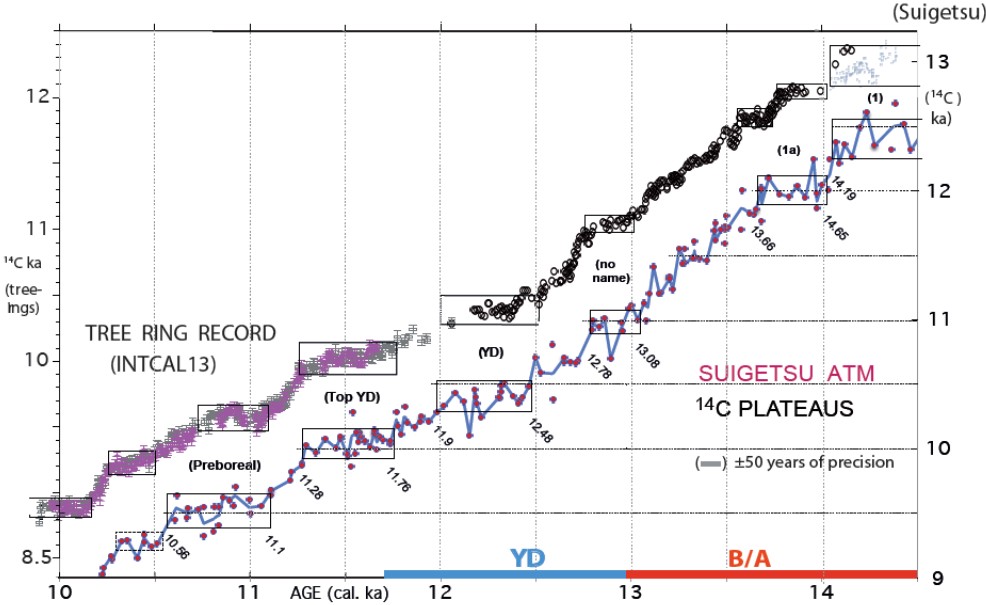

Figure 3. (Courtesy of Bronk Ramsey et al., 2020). A comparison of Bayesian spline compilations
of datasets of Suigetsu (green) and Hulu speleothem (magenta) over the period 20-35 cal. ka,
transformed using a MatLab deconvolution algorithm (linear ramp with mean of 420 years).
Gradually decreasing atmospheric $\Delta^{14}$C values reflect atmospheric [14]C age plateaus and their
uncertainty range (black numbers in brackets). Rising $\Delta^{14}$C values reflect atmospheric [14]C age
jumps.



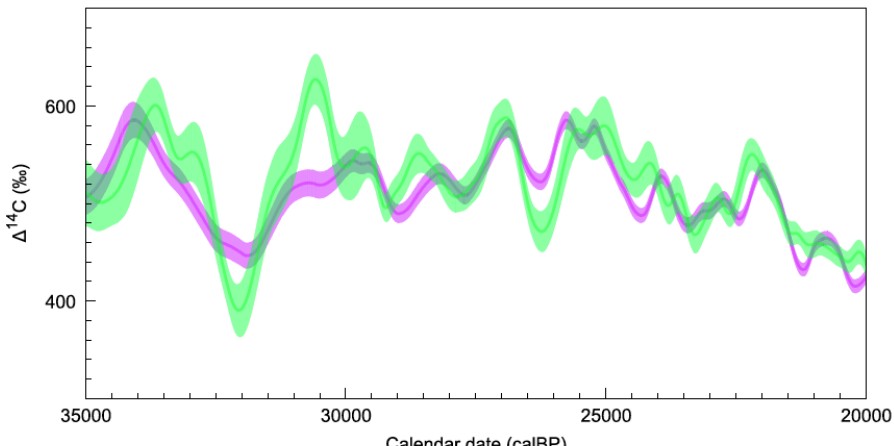



Figure 4. Atmospheric [14]C ages and error bars of Lake Suigetsu plant macrofossils vs. U/Th-based
model age of 15–21 (bottom) and 21–27 (top) cal. ka (blue dots; Sarnthein et al., 2020, modified,
using age data of Bronk Ramsey et al., 2020). [14]C plateaus longer than 250 yr are outlined by a
suite of labeled horizontal boxes that envelop scatter bands of largely constant or slightly rising
[14]C ages, separated by "[14]C jumps". Red boxes outline contemporary [14]C plateaus found in the
Hulu [14]C record. 1:1 line reflects a gradient of one [14]C yr per cal. yr.



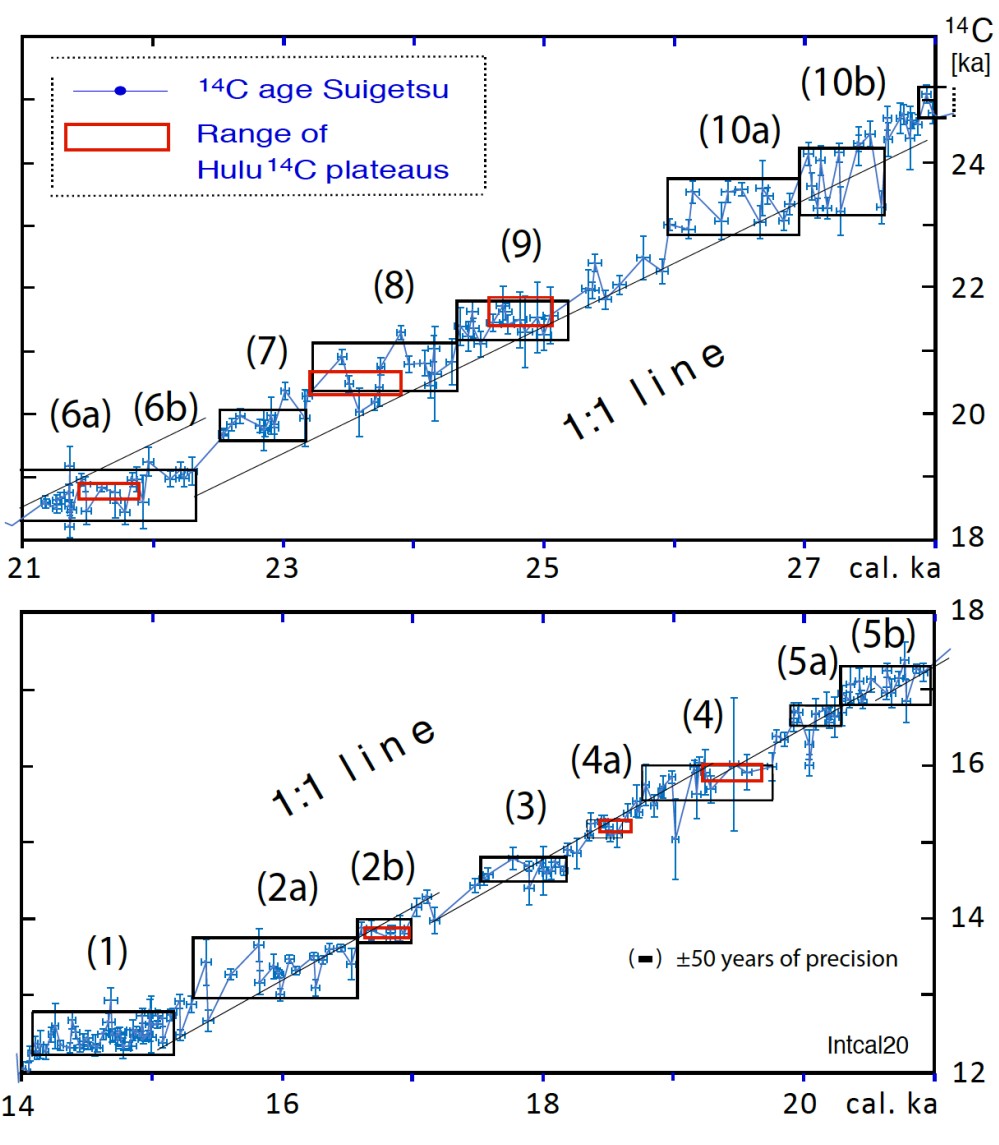





***14C PLATEAU TUNING – A MISLEADING APPROACH OR TRENDSETTING TOOL FOR MARINE***
***PALEOCLIMATE STUDIES?***

*Michael Sarnthein[1] and Pieter M. Grootes[2]*

**SUPPLEMENTARY MATERIALS** *(i.e., just an overview for a fast-reading reviewer)*

SUPPLEMENTARY FIGURES AND TABLES
Figures S1 - S19.
Planktic $^{14}$C records of sediment cores plotted vs. core depth. Core location and references to
data source are given in Table S20. Planktic $^{14}$C plateaus (horizontal boxes) are compared to
atmospheric (atm) $^{14}$C plateau suite of Lake Suigetsu (Bronk Ramsey et al., 2020), where
calendar ages of plateau boundaries (and average atmospheric $^{14}$C ages; Fig. S12) are given
below. Local planktic reservoir ages (in blue) result from the difference between the average raw
$^{14}$C age of planktic $^{14}$C plateaus measured in the core and the $^{14}$C age of equivalent atmospheric
$^{14}$C plateaus numbered 1 – 10 (numbers in brackets). Top panel shows units of the 1$^{st}$ derivative
($^{14}$C yr per m core depth) and 1-$\sigma$ uncertainty range, with high values indicating $^{14}$C jumps and
$^{14}$C plateaus (numbered in red) constrained at 'half-height' by asterisks (as defined in Sarnthein
et al. 2015). B/A = Bølling-Allerød; HS-1, HS-2 = Heinrich Stadial 1 and Heinrich Stadial 2; LGM =
Last Glacial Maximum. Sedimentation rates are based on ages of $^{14}$C plateau boundaries. Red
double slash indicates sedimentation gap.

Suppl. Tables S1 - S19.
Planktic and benthic $^{14}$C ages measured in 19 ocean sediment cores. All cal. ages (yr BP) were
deduced by means of $^{14}$C plateau tuning and adjusted to the IntCal20 U/Th-based model time





scale of Bronk Ramsey et al. (2020). Core locations and data sources are listed in Table S20.
Tables S1 - S19 are being deposited at Pangaea.de.

Suppl. Table S 20. Core locations and data sources (and references) for 19 core sites from key
positions in the world ocean, used for generating a PT-based time scale for planktic and benthic
$^{14}$C ages displayed in Tables S1 - S19.
==========================================================