# Peer review of "Michael Sarnthein[1] and Pieter M. Grootes[2]"

_Climate of the Past, 2021_

## Community Comment (CC1)

**$^{14}$C plateau tuning – A misleading approach for marine paleoclimate studies**

Edouard Bard[1] and Timothy J. Heaton[2]

1 CEREGE, Aix-Marseille University, CNRS, IRD, INRAE, Collège de France,
Technopôle de l'Arbois, Aix-en-Provence, France

2 School of Mathematics and Statistics, University of Sheffield, Sheffield S3 7RH, UK

We have already published, in 2021, a paper in CP (Bard & Heaton, 2021) which identified and evaluated the pitfalls of the Plateau Tuning (PT) method. In this previous paper, we expressed significant doubts regarding the ability of PT to simultaneously **1/** construct the chronologies of deep-sea sediment cores; and **2/** obtain marine reservoir ages (MRA) in a robust or precise manner.

Unfortunately, in their new paper, Sarnthein & Grootes do not address our criticisms with any substantial new work, i.e., by provision of new data, independent verification of the results they infer using PT, or statistical calculation. We therefore refer the reader to our original 2021 paper for a detailed point-by-point discussion of PT's issues (see also our responses to Sarnthein & Grootes and Grootes & Sarnthein' $\approx$ 50 pages of Community Comments which accompany this 2021 paper). Readers can also consult a recent review paper about radiocarbon in paleoceanography by Skinner & Bard (2022) which provides examples of PT results that lead to large and unexplained discrepancies with other records from the same areas. We feel it is not helpful to repeat all the same arguments here.

In this new submission, Sarnthein & Grootes only consider one narrow aspect of PT – whether long atmospheric $^{14}$C-age plateaus exist and can be reliably identified. We are unfortunately still unconvinced that such an extreme atmospheric $^{14}$C-age step-ladder is physically plausible or supported by observational data. The authors do not consider concerns regarding other key PT requirements: whether these atmospheric $^{14}$C-age plateaus transfer to the marine environment; and, even if they do, whether they can be reliably identified in extremely sparse $^{14}$C samples from sediment cores for which no independent timescale is available and for which the MRA correction is a priori variable.

We are therefore afraid that we remain of the view that PT cannot be seen as a reliable approach to the building of chronologies for marine sediment cores, and the results presented here on the changes in MRA of their plateau-tuned cores must be viewed with caution.

**What Sarnthein and Grootes claim in their new work**

Sarnthein & Grootes argue that they have proved the authenticity of their hypothesized atmospheric $^{14}$C-age plateaus based on eyeballed comparisons with $^{14}$C measurements from tree-rings and Hulu Cave. However, as we explain below, these comparisons involve considerable circularity (and we remain unclear how a reader can be confident that the hypothetical $^{14}$C-age plateaus have been identified objectively). We therefore strongly believe that no objective statement can be made about the authenticity of the hypothesized atmospheric $^{14}$C-age plateaus.

In the new submission, Sarnthein & Grootes have revised their 2020 suite of atmospheric $^{14}$C-age plateaus – removing/merging some, while editing the starts and ends of others. These

updated, hypothesized, atmospheric $^{14}$C-age plateaus remain based upon the $^{14}$C record of Lake Suigetsu, but with its revised calendar age chronology that was published by Bronk Ramsey et al. (2020).

Based upon the new hypothesized atmospheric $^{14}$C-age plateaus, Sarnthein & Grootes revise the chronologies and MRA records of 19 deep-sea cores which they have published over the last decade. Having used the same 19 core datasets, and the same PT method, it is no surprise that the new results are very similar to what they had before. This new paper therefore provides limited new insight over their review published in 2020 in CP (Sarnthein et al. 2020).

**Lack of independence between Lake Suigetsu and Hulu Cave Chronologies**

Critically, readers need to be aware that the updated Suigetsu calendar age chronology (on which the new hypothesized $^{14}$C-age plateaus are based) is not independent of the Hulu Cave $^{14}$C record. Indeed, to create the updated Lake Suigetsu calendar age chronology, the (independent) varve-ages were adjusted/tuned so that the overall Suigetsu $^{14}$C vs cal age record better matched the $^{14}$C vs cal age record from the U-Th dated Hulu Cave stalagmites that constitute the backbone of the IntCal20 calibration curve (Reimer et al., 2020; Heaton et al., 2020a, 2021). See Cheng et al. (2018) for details of the Hulu $^{14}$C vs U-Th record; and Bronk Ramsey et al. (2020) for a description of the Lake Suigetsu tuning that adjusts the varve counts of Schlolaut et al. (2018).

The (Hulu-tuned) Lake Suigetsu calendar age timescale differs by up to 3 cal kyrs from the original varved timescale. In addition to adjusting the long-term trend, the tuning also concerns the millennial-scale structures (see Figs. 1 and 5 by Bronk Ramsey et al. (2020); this Fig. 5 being reprinted as Fig. 3 by Sarnthein & Grootes). Due to this tuning, it is to be expected that the Hulu and Lake Suigetsu $^{14}$C records show similar features at similar times – however, due to the lack of independence between the timescales of the records, and that $^{14}$C features are themselves used in the tuning, one must be extremely careful not to overstate one's confidence that similarities in the two records after tuning provide robust and repeated evidence for the simultaneous presence of $^{14}$C-age plateaus.

Until we can find an alternative chronology, then inference from Lake Suigetsu which uses its (Hulu $^{14}$C-informed) calendar age information is not *solely* atmospheric. The tying to Hulu $^{14}$C transfers many of the complexities of Hulu Cave to Lake Suigetsu - in particular, the assumption that the $^{14}$C depletion in Hulu is equivalent to a constant dead carbon fraction.

Any $^{14}$C-age plateaus identified using the Lake Suigetsu $^{14}$C record are not therefore atmospheric-only. Furthermore, at least part of the agreement with Hulu Cave is due to (or at least reinforced by) the tuning which creates the Suigetsu timescale.

**Variation in Atmospheric $^{14}$C levels**

Sarnthein & Grootes continue to confuse the question of whether there is evidence for a long staircase of $^{14}$C-age plateaus that can be identified based upon Lake Suigetsu; with the question of whether there is high-frequency variation in atmospheric $^{14}$C levels from 30 – 15 cal kyr BP.

As explained in Reimer et al. (2020), Heaton et al. (2020a) and Bard & Heaton (2021), it is highly likely that there is currently unknown variation in atmospheric $^{14}$C levels from 30 – 15 cal kyr BP. Variation in atmospheric $^{14}$C levels is seen from 15 – 0 cal kyr where we have directly-atmospheric tree rings and we would expect similar variations (at least) to have occurred beforehand. However, this does not mean either that:

- the structure from 30 – 15 cal kyr BP takes the form of long $^{14}$C-age plateaus separated by huge jumps in atmospheric $^{14}$C levels (i.e., implying that $^{14}$C ages stay constant through time during 80% of the 30 – 14 cal kyr BP interval); or
- we can reliably and precisely separate genuine atmospheric $^{14}$C variation from random fluctuations using the Lake Suigetsu $^{14}$C record.

and it is misleading to conflate these issues.

[Figure]

***Figure 1:*** *Plot of the IntCal20 curve (during the time period where it is based upon gold-standard tree-ring $^{14}$C determinations) against the hypothesized (2022) $^{14}$C-age plateaus of Sarnthein & Grootes. Shown as small, light grey, dots are the underlying tree ring determinations. Evidence for the hypothesized plateaus amongst the tree ring determinations is unclear – arguments of equal strength could potentially be made that several other time periods exhibit similar tree-ring behavior as those identified as plateau periods by Sarnthein & Grootes. Equally, questions could be raised as to the length of the plateaus that have been identified.*

For the period covered by $^{14}$C data on tree-rings, there are no $^{14}$C-age plateaus that extend over 1000 cal yrs (or longer) as hypothesized by Sarnthein & Grootes for the older part of the record. Further, Sarnthein & Grootes hypothesize a set of $^{14}$C-age plateaus from 14 – 10 cal kyr BP based upon Lake Suigetsu. These are shown in Figure 1 alongside the IntCal20 curve, and its constituent *gold-standard* tree-ring measurements. Sarnthein & Grootes argue that the $^{14}$C-age plateaus they identify in this period agree with *gold-standard* tree-ring $^{14}$C determinations. However, as can be seen in Figure 1, there is considerable variation in the $^{14}$C-ages of tree-rings throughout this entire 4 cal kyr period with a range of inversions and wiggles and no

objective definition of what should be considered to constitute a tree-ring plateau is given. To our eyes, the tree-rings present no obvious evidence for the $^{14}$C-age plateaus they propose for tuning from 11.1 – 10.6 cal kyr BP; 11.8 – 11.3 cal kyr BP; and from 13.1 – 12.8 cal kyr BP; or at least no stronger plateau behavior than other directly neighboring time periods. This shows the difficulty of relying upon Lake Suigetsu alone to identify atmospheric variation. Further, without strong plateau behavior, the question of how to reliably identify matching $^{14}$C features in sparse marine records remains.

The only $^{14}$C-age plateau proposed in their suite which does appear to be seen in the tree rings might be from 12.5 – 11.9 cal kyr BP, however even this appears to be shorter in length, perhaps lasting only 300 cal yrs in the tree ring records. Even plateau 1A (Bölling-Alleröd) is not really a plateau when one incorporates the recent floating tree-ring sequences by Adolphi et al. (2017) – see Heaton et al. (2020a) and Muscheler et al. (2020).

**Updating of hypothesized $^{14}$C-age plateaus**

The Figure 1 by Sarnthein & Grootes shows the resulting $\Delta^{14}$C for the 30 – 14 cal kyr BP in which $^{14}$C-age plateaus appear as decreasing trends equivalent to the radioactive decay (1‰ over 8 years). This graph is an update of Fig. 3a in our 2021 paper in CP. On this Figure, Sarnthein & Grootes compare their new plateau record (in bright purple) with the records we calculated with their previous definitions of plateaus.

The main changes in Fig. 1 by Sarnthein & Grootes are that former $^{14}$C-age plateaus 6a and 6b have been merged and that plateau 7 has disappeared. In our Figure 2 below, we have remade Sarnthein & Grootes' submitted figure since the lines shown in their original do not appear to correspond to the calendar ages of the plateaus as given in their paper, notably for plateau 7 and the end of plateau 10a. Our Figure 2 is based on Table 1 of Sarnthein & Grootes (2022) which seems incompatible with their Figure 1. In blue, we show the implied $\Delta^{14}$C record based on the 2022 hypothesized $^{14}$C-age plateaus using the updated Lake Suigetsu record, in green the implied $\Delta^{14}$C using the $^{14}$C-age plateaus proposed in 2020.

Our 2021 CP paper was actually the first to provide a $\Delta^{14}$C plot illustrating the implications of the atmospheric $^{14}$C-age plateaus hypothesized by Sarnthein & Grootes. Our original intention was to show how physically unreasonable the atmospheric $\Delta^{14}$C record must be in order to allow for such $^{14}$C-age plateaus (a succession of very abrupt jumps followed by ramps cancelling the radioactive decay).

It remains the case that no mechanism is proposed to explain these massive and instantaneous $\Delta^{14}$C increases (ca. 100 to 200 ‰) that occur without any relationship with $^{10}$Be maxima in ice cores. It is thus surprising that in the summary of their new paper Sarnthein & Grootes claim the discovery of *"fine structure of jumps and plateaus in atmospheric and planktic radiocarbon ($^{14}$C) concentration that reflect authentic changes in atmospheric $^{14}$C production."*

[Figure]

**Figure 2:** *Inferred Δ14C reconstructions based upon the hypothesized 14C-age plateaus proposed by Sarnthein & Grootes in 2020 (green) and here in 2022 (blue). Panel (a) shows the plateau-based Δ14C reconstructions plotted against the Lake Suigetsu 14C-determinations and a Suigetsu-only calibration curve. Panel (b) shows the plateau-based reconstructions against the IntCal20 Δ14C estimate. Note: These plots are somewhat different from Figure 1 by Sarnthein & Grootes since their plot is not consistent with the calendar dates they provide (in their Table 1) for the hypothesized 14C-age plateaus.*

**Transferal of atmospheric 14C variations to the ocean**

Besides redoing PT tuning on their 19 ocean sediment records, Sarnthein & Grootes have not provided any new data or calculations to answer criticisms expressed in our 2021 paper.

In particular, none of our statistical concerns have been taken into account. The level of objectivity in the identification of plateaus in marine sediments remains unclear, and we have doubts as to how reproducible this identification process is when performed by others. Further, the authors provide very little discussion regarding the sparsity of most marine 14C records, and how this likely prevents identification of any short-term and fine scale structure – especially when such sediment records do not have a timescale of their own and structure is only seen against depth (which, with PT, has a highly non-linear relationship with calendar age). In addition, the MRA is also variable through time which further complicates the tuning.

No independent evidence is provided to verify the inferred extreme changes of sedimentation rate, including hiatuses. Ground-truthing of PT should come from an independent comparison between PT and a detailed core chronology (based on other independent techniques) to derive sedimentation rates and reservoir age changes. This is a prerequisite to demonstrate the merit

and added-value of PT. Such ground-truthing would also likely require objective criteria to identify the plateaus.

As shown by Fig. S1 to S19 by Sarnthein & Grootes, the variations in the sedimentation rate inferred by PT show very large downcore changes – ranging from several-fold to an order of magnitude (and infinity for hiatuses). These downcore changes (specifically the minimum sedimentation rate within any core) is what matters for signal alteration by bioturbation and other sediment mixing processes (i.e., not the mean sedimentation rate over the full core). Foraminifera counts are still not provided for the 19 cores and no effort has been made to model the effect of sediment mixing.

Sarnthein & Grootes still cite the results on four South Pacific cores with a reference to a paper by Küssner et al. (2020), which we understand has been rejected for publication in *Paleoceanography & Paleoclimatology* (cited as submitted to this journal in Sarnthein et al. 2020). These are the records which were significantly changed between the submitted and published versions of Sarnthein et al. (2020) without explanation (see the submitted version available publicly from the CP web site). Furthermore, two independent Community Comments accompanied our 2021 paper (Lamy & Arz 2021, Michel & Siani 2021), confirming our initial doubts about PT applied to those cores (Drs. E. Michel and G. Siani are listed as coauthors of the paper by Küssner et al., Dr. F. Lamy was listed as a coauthor of Küssner et al. in the submitted version by Sarnthein et al. 2020). In the new paper by Sarnthein & Grootes, the PT results on those South Pacific cores are now cited as a PANGAEA database by Küssner et al. (2020). This is clearly not satisfactory as PT relies heavily on subjective interpretations, which should be described and provided in a peer-reviewed paper.

Sarnthein & Grootes still think that the marine $^{14}$C records can be matched directly to the atmospheric record, without taking into account the smoothing and lagging effects of carbon uptake and mixing in the surface ocean. The level of such oceanic smoothing is directly related to the value of the surface reservoir age. This adds inextricable complexity to PT supposed to provide both the chronology of the core and the local MRA record at the core site (with MRAs that are often up to several millennia inferred by PT). Our carbon cycle modeling results clearly illustrated this point (Bard & Heaton 2021). They were obtained with a rather generic 12-box model which had been compared with other models and with the more realistic 2D Bern model (Delaygue & Bard 2011). Such simple box models have been used in the frame of the IntCal group for decades (e.g., Stuiver & Braziunas 1993, Reimer et al. 2013, Heaton et al. 2020b).

Besides the multiple anomalies already mentioned in previous papers (Bard & Heaton 2021, Skinner & Bard 2022), the new submission by Sarnthein & Grootes led us to think about the pair of cores located in the South China Sea (GIK17940 and SO50-37, Figs. S6 and S7, respectively). The PT chronologies based on planktic $^{14}$C ages lead to large MRA values in both cores ranging between 900 and 1900 $^{14}$C yr. Surprisingly, the benthic $^{14}$C ages in GIK17940 are about the same as planktic $^{14}$C ages (Fig. S6), which leads to the conclusion that the water column was old, but completely mixed down to 2 km (the depth of the core is 1721 m). By contrast, the nearby core SO50-37, collected somewhat deeper (2655 m), exhibits $^{14}$C ages on benthic foraminifera that are about 2-3 kyr older than planktics for the same time periods (Fig. S7). Given the deep homogenization invoked by Sarnthein & Grootes for that site (down to 2km), the benthic $^{14}$C record of GIK17940 should also show the hypothesized age plateaus, which is obviously not the case (Fig. S6). This puzzling example is probably spurious and is not reassuring for the reliability of PT.

**References**

Adolphi, F., Muscheler, R., Friedrich, M., Güttler, D., Wacker, L., Talamo, S., and Kromer, B.: Radiocarbon calibration uncertainties during the last deglaciation: Insights from new floating tree-ring chronologies, *Quat. Sci. Rev.*, 170, 98–108, https://doi.org/10.1016/j.quascirev.2017.06.026, 2017.

Bard, E. and Heaton, T. J.: On the tuning of plateaus in atmospheric and oceanic $^{14}C$ records to derive calendar chronologies of deep-sea cores and records of $^{14}C$ marine reservoir age changes, *Clim. Past*, 17, 1701–1725, https://doi.org/10.5194/cp-17-1701-2021, 2021.

Bronk Ramsey, C., Heaton, T. J., Schlolaut, G., Staff, R. A., Bryant, C. L., Brauer, A., Lamb, H. F., Marshall, M. H., and Nakagawa, T.: Reanalysis of the atmospheric radiocarbon calibration record from Lake Suigetsu, Japan, *Radiocarbon*, 62, 989–999, https://doi.org/10.1017/RDC.2020.18, 2020.

Cheng, H., Lawrence Edwards, R., Southon, J., Matsumoto, K., Feinberg, J. M., Sinha, A., Zhou, W., Li, H., Li, X., Xu, Y., Chen, S., Tan, M., Wang, Q., Wang, Y., and Ning, Y.: Atmospheric $^{14}C/^{12}C$ changes during the last glacial period from hulu cave, *Science*, 362, 1293–1297, https://doi.org/10.1126/science.aau0747, 2018.

Delaygue, G. and Bard, E.: An Antarctic view of beryllium-10 and solar activity for the past millennium, *Clim. Dynam.*, 36, 2201–2218, https://doi.org/10.1007/s00382-010-0795-1, 2011.

Grootes, P. M. and Sarnthein, M.: Community comment on "On the tuning of plateaus in atmospheric and oceanic $^{14}C$ records to derive calendar chronologies of deep-sea cores and records of $^{14}C$ marine reservoir age changes" by E. Bard and T. J. Heaton, *Clim. Past Discuss.*, https://doi.org/10.5194/cp-2020-164-CC2, 2021.

Heaton, T. J., Blaauw, M., Blackwell, P. G., Bronk Ramsey, C., Reimer, P. J., and Scott, E. M.: The IntCal20 approach to radiocarbon calibration curve construction: A new methodology using bayesian splines and errors-in-variables, *Radiocarbon*, 62, 821–863, https://doi.org/10.1017/RDC.2020.46, 2020a.

Heaton, T. J., Köhler, P., Butzin, M., Bard, E., Reimer, R. W., Austin, W. E. N., Bronk Ramsey, C., Grootes, P. M., Hughen, K. A., Kromer, B., Reimer, P. J., Adkins, J., Burke, A., Cook, M. S., Olsen, J., and Skinner, L. C.: Marine20—The marine radiocarbon age calibration curve (0–55,000 cal BP), *Radiocarbon*, 62, 779–820, https://doi.org/10.1017/RDC.2020.68, 2020b.

Heaton, T. J., Bard, E., Bronk Ramsey, C., Butzin, M., Köhler, P., Muscheler, R., Reimer, P. J., and Wacker, L.: Radiocarbon: A key tracer for studying Earth's dynamo, climate system, carbon cycle, and Sun, *Science*, 374, eabd7096, https://doi.org/10.1126/science.abd7096, 2021.

Küssner, K., Sarnthein, M., Michel, E., Mollenhauer, G., Siani G., and Tiedemann, R.: Glacial-to-deglacial reservoir ages of surface waters in the southern South Pacific. PANGAEA, https://doi.org/10.1594/PANGAEA.922671, 2020.

Lamy, F. and Arz, H. W.: Community comment on "On the tuning of plateaus in atmospheric and oceanic $^{14}C$ records to derive calendar chronologies of deep-sea cores and records of $^{14}C$ marine reservoir age changes" by E. Bard and T. J. Heaton, *Clim. Past Discuss.*, https://doi.org/10.5194/cp-2020-164-CC5, 2021.

Michel, E. and Siani, G.: Community comment on "On the tuning of plateaus in atmospheric and oceanic $^{14}C$ records to derive calendar chronologies of deep-sea cores and records of $^{14}C$

marine reservoir age changes" by E. Bard and T. J. Heaton, *Clim. Past Discuss.*, https://doi.org/10.5194/cp-2020-164-CC6, 2021.

Muscheler, R., Adolphi, F., Heaton, T. J., Bronk Ramsey, C., Svensson, A., van der Plicht, J., and Reimer, P. J.: Testing and improving the IntCal20 calibration curve with independent records, *Radiocarbon*, 62, 1079–1094, https://doi.org/10.1017/RDC.2020.54, 2020.

Reimer, P. J., Bard, E., Bayliss, A., Beck, J. W., Blackwell, P. G., Bronk Ramsey, C., Buck, C. E., Cheng, H., Edwards, R. L., Friedrich, M., Grootes, P. M., Guilderson, T. P., Haflidason, H., Hajdas, I., Hatté, C., Heaton, T. J., Hoffmann, D. L., Hogg, A. G., Hughen, K. A., Kaiser, K. F., Kromer, B., Manning, S. W., Niu, M., Reimer, R. W., Richards, D. A., Scott, E. M., Southon, J. R., Staff, R. A., Turney, C. S. M., and van der Plicht, J.: IntCal13 and Marine13 radiocarbon age calibration curves 0–50 000 Years cal BP, *Radiocarbon*, 55, 1869–1887, https://doi.org/10.2458/azu_js_rc.55.16947, 2013.

Reimer, P. J., Austin, W. E. N., Bard, E., Bayliss, A., Blackwell, P. G., Bronk Ramsey, C., Butzin, M., Cheng, H., Edwards, R. L., Friedrich, M., Grootes, P. M., Guilderson, T. P., Hajdas, I., Heaton, T. J., Hogg, A. G., Hughen, K. A., Kromer, B., Manning, S. W., Muscheler, R., Palmer, J. G., Pearson, C., Van Der Plicht, J., Reimer, R. W., Richards, D. A., Scott, E. M., Southon, J. R., Turney, C. S. M., Wacker, L., Adolphi, F., Büntgen, U., Capano, M., Fahrni, S. M., Fogtmann-Schulz, A., Friedrich, R., Köhler, P., Kudsk, S., Miyake, F., Olsen, J., Reinig, F., Sakamoto, M., Sookdeo, A., and Talamo, S.: The IntCal20 northern hemisphere radiocarbon age calibration curve (0–55 cal kBP), *Radiocarbon*, 62, 725–757, https://doi.org/10.1017/RDC.2020.41, 2020.

Sarnthein, M., Küssner, K., Grootes, P.M., Ausin, B., Eglinton, T., Muglia, J., Muscheler, R., Schlolaut, G. Plateaus and jumps in the atmospheric radiocarbon record – Potential origin and value as global age markers for glacial-to-deglacial paleoceanography, a synthesis. *Climate of the Past,* 16, 2547–2571, https://doi.org/10.5194/cp-16-2547-2020, 2020

Sarnthein, M. and Grootes, P. M.: Community comment on "On the tuning of plateaus in atmospheric and oceanic $^{14}$C records to derive calendar chronologies of deep-sea cores and records of $^{14}$C marine reservoir age changes" by E. Bard and T. J. Heaton, *Clim. Past Discuss.*, https://doi.org/10.5194/cp-2020-164-CC1, 2021.

Sarnthein, M. and Grootes, P. M.: $^{14}$C plateau tuning – A misleading approach or trendsetting tool for marine paleoclimate studies?, *Clim. Past Discuss.*, https://doi.org/10.5194/cp-2021-173, 2022.

Schlolaut, G., Staff, R. A., Brauer, A., Lamb, H. F., Marshall, M. H., Bronk Ramsey, C., and Nakagawa, T.: An extended and revised Lake Suigetsu varve chronology from ∼50 to ∼10 ka BP based on detailed sediment micro-facies analyses, *Quat. Sci. Rev.*, 200, 351–366, https://doi.org/10.1016/j.quascirev.2018.09.021, 2018.

Skinner, L. and Bard, E.: Radiocarbon as a dating tool and tracer in paleoceanography, *Rev. Geophys.*, 60, 1, 1–64, https://doi.org/10.1029/2020RG000720, 2022.

Stuiver, M., and Braziunas, T. Modeling atmospheric $^{14}$C influences and $^{14}$C ages of marine samples to 10,000 BC. *Radiocarbon, 35*(1), 137-189, https://doi.org/10.1017/S0033822200013874, 1993.

---

## Community Comment (CC2)

Discussion contribution Climate of the Past.
['Comment on cp-2021-173'](), by Edouard Bard and Timothy J. Heaton.
**[14]C plateau tuning – A misleading approach for marine paleoclimate studies.**

First reply and invitation for further clarification by Pieter M. Grootes

We (Sarnthein and Grootes) welcome the discussion contribution of Bard and Heaton (B&H) to '[14]C Plateau Tuning – A misleading approach or a trendsetting tool for marine paleoclimate studies?'. This continues the evaluation, started with Sarnthein et al 2020, of Plateau Tuning (PT) as a paleoceanographic research tool to evaluate small [14]C signals in a patchy, noisy record where traditional methods fail.

We submitted to Climate of the Past (CP) to provide a summary of our PT results for 19 cores and to invite community discussion of the new PT technique and its, for some, sometimes controversial results. After considerable discussion, the paper was published late 2020. B&H submitted comments with serious objections in late January 2020 - yet listed as received by CP on 23 December 2020 - after Sarnthein et al 2020 had been published (Bard and Heaton, CP 2021, p.1701-1725). B&H raised 17 objections spread over two chapters. As pointed out in our 2021 CP comments, it seems 'Its aim is to demonstrate that Plateau Tuning (PT) is fraught with problems and should not be used'. The present B&H comment appears to follow this line.

B&H start by referring to objections in their long 2021 CP-paper and the detailed discussion of those objections in our comments and their answers. We answered in 2021 in detail B&H objections related to the difficulty of identifying plateaus and to the lack of statistical robustness (in 2.3, 3.2, 3.4, 3.7, and 3.8). B&H selectively grouped comments and ignored several, both in their rebuttal and in the final paper.
Our present paper is the promised update of Sarnthein et al., 2020 that brings together all our data sets using the new Suigetsu time scale of Bronk Ramsey et al 2020, showing the information that may be obtained from 'difficult' sediment records by using Plateau tuning. To answer the question in its title we need to consider, in addition to the present B&H comments, some earlier discussion points that B&H failed to address in CP 2021.For a meaningful discussion I invite B&H's further response to the points listed below.

**The difference between PT and IntCal20:**
**PT** aims to extract new information regarding variations in internal ocean dynamics and ocean-atmosphere exchange over last glacial-deglacial-Holocene times from ocean sites that lack clear chronostratigraphic markers for detailed age control, e.g., as listed in 'Outlook' of B&H 2021. PT hunts for small [14]C signals in a patchy, noisy record that will generally fail statistical tests for robustness. This is even true for the fine structure of the Suigetsu atmospheric [14]C record.
**IntCal20**, by contrast, is a statistically robust tool to translate [14]C ages into calendar ages. To achieve robustness, decadal to centennial information has been smoothed, which makes it less suitable for PT, as illustrated by Fig. 2 of the present B&H comment, comparing the Bayesian splines of Suigetsu and IntCal20.

**PT robustness.**

As pointed out earlier: Absence of statistical proof is not proof of absence. The lack of statistical robustness of a single PT $^{14}$C record, emphasized in the statistical perspective of B&H 2021, is compensated in PT by much work, that documents consistency of the derived $^{14}$C plateau sequence with local sedimentology, stratigraphy, and the multi-parameter sediment record, and with similar plateau sequences developed elsewhere. The probability that a specific $^{14}$C fluctuation is caused by noise decreases with each new record in which this fluctuation is found, thus building robustness. The set of 19 records provides a base to check the global consistency.

**Incorrect PT assumptions B&H 2021 used in formulating their objections.**

Bard is an experienced paleoceanographer and his list of potential problems of sediment records is realistic. Yet, contrary to his writing, these problems have been recognized by Sarnthein and coworkers in developing PT. In our comments 2021 and in the manuscript presently submitted, we pointed out that B&H's objections were based on their misunderstanding of PT and that several of their supposed 'assumptions made in PT' were incorrect.

**Physical impossibility of PT plateau schedule**

B&H discussed (in 2.3) in great detail the timing of $^{14}$C fluctuations in the PT-tuned Suigetsu record. They objected that PT tuning resulted in a highly irregular behaviour of the $^{14}$C clock and, when moved from $^{14}$C age to $^{14}$C concentration $\Delta^{14}$C, in physically unrealistic jumps in atmospheric $^{14}$C concentration. B&H repeat this objection in the present comment.
They ignore our comments to 2.3 that the jumps were the result of our use of constant-age plateaus that was dictated by the lack of data sufficient to define a plateau slope. An irregular $^{14}$C clock, moreover, is the logical consequence of fluctuating atmospheric $^{14}$C concentrations.

**Bayesian spline and age plateaus.**

B&H made the Bayesian spline plot of Suigetsu $\Delta^{14}$C values to demonstrate the physically unreasonable consequences of PT. Despite our comments, B&H so far do not acknowledge the surprising agreement between their Bayesian spline and the PT record of atmospheric plateaus and jumps, especially after the recent time scale update. Figure 2 of their present comment - unfortunately including a typo in our Table 1; thanks to B&H for finding it – again demonstrates this. Fig. 2 also shows the loss of fine structure in IntCal20.

---

## Author Comment (AC2)

Dr. Michael Sarnthein, Prof. emer.                                    26 April 2022
Institut für Geowissenschaften
University of Kiel
Olshausenstr. 40
24098  K i e l,  Germany
<michael.sarnthein@ifg.uni-kiel.de>

**COMMENTS to Review #1**

by Michael Sarnthein and Pieter M. Grootes (authors)

The format and discussion of this document better resembles a comment than a scientific manuscript, making this very difficult to understand / review. I can only guess that the authors intended for this text to be submitted as a comment instead of a stand-alone manuscript. It is therefore simplest to reject this submission and allow the authors to correct their mistake or ask them to submit a manuscript that at least partially follows the standard formatting of a scientific publication.

A brief listing of the problems with this document.
The title of this document suggests the reader will be provided with an expert review of the Plateau tuning technique for identifying the calendar age. However, the Introduction does little to introduce the reader to the relevant argument, instead directing the reader to first fully understand the discussions of several earlier publications. This is a missed opportunity.

Indeed, our text in part may have been misled by following too excessively the concept to keep the text as short as possible by referring to the comprehensive review of Plateau Tuning (PT) technique already given in our synthesis paper of December 2020. We see no problem now to insert a proper summary on PT into the Introduction of this manuscript.

The reviewer is correct in his guess that the manuscript in part is a comment in the ongoing discussion regarding the potential of Plateau Tuning. As such the title may be modified to *"$^{14}$C plateau tuning – A tool for marine paleoclimate studies?"*, which might be more appropriate as the paper presents new evidence for the applicability of PT as summarized in the abstract. For the sake of brevity, we largely refer to the extensive discussion of the arguments Pro and Contra PT of Sarnthein et al, 2020, and Bard&Heaton, 2021.

If the rebuttal to this comment is that the review was already published, then I see no reason why the text provided here should be a stand-alone manuscript. Submit it as a comment to the already peer-reviewed manuscripts.

Crucial is that transfer of the Suigetsu atmospheric $^{14}$C data to the new Bronk Ramsey et al. 2020-time scale provides a new, revised pattern of atmospheric $^{14}$C 'plateaus' and 'jumps'. The present manuscript, however, is centered on four distinct targets beyond a rebuttal of allegations, that are **(1)** to display the authenticity of the atmospheric $^{14}$C plateau pattern on the basis of a close match between results of a Bayesian spline of Suigetsu $\Delta^{14}$C/age data (provided by Bard & Heaton, 2021, but similarity not mentioned) and those deduced by means of both visual inspection and 1st derivative method in PT, **(2)** to document the global significance of the Suigetsu atmospheric $^{14}$C record by its coherence with the tree ring record 10 to ~15 cal. ka, **(3)** to adjust the detailed cal.-age chronology to the age control recently published by Bronk Ramsey et al., a correction crucial for all global high-resolution age correlations of ocean sediment cores, and **(4)** to provide detailed arguments showing that sediment distortions by differential bioturbational mixing do not form any major origin of "fake" $^{14}$C plateaus.

We regard it as legitimate to shortly publish on these items 'discovered' in the B&H 2021 discussion subsequent to our synthesis paper of Dec. 2020.

No Results section or Discussion is provided, with the text immediately moving to arguments that I (a putative expert) have difficulty following.

As said earlier, the manuscript is a comment in the ongoing discussion regarding the potential of Plateau Tuning. There is no problem, however, to provide a better structure of the manuscript by adding additional headlines for pertinent paragraphs to meet the standard formatting of a scientific publication postulated. 'Cautionary' comments against this manuscript submitted by B&H to CP Discussions potentially ask for additional changes/additions to the text anyway.

Every figure has a different X-axis range, with some have the calendar age move in different directions. These figures are not professional.

We feel sorry for this mistake. The differing X-axis of Figure 3 has now been inverted to match the sense of figures 1, 2, and 4.

REFERENCES

Bard, E. and Heaton, T.J.: On the tuning of plateaus in atmospheric and oceanic 14C records to derive calendar chronologies of deep-sea cores and records of 14C marine reservoir age changes. Climate of the Past, 17, 1701–1725. https://doi.org/10.5194/cp-17-1701-2021, 2021.

Bronk Ramsey, C., Heaton, T.J., Schlolaut, G., Staff, R.A., Bryant, C.L., Lamb, H.F., Marshall, M.H., Nakagawa, T.: Reanalysis of the atmospheric radiocarbon calibration record from Lake Suigetsu, Japan. Radiocarbon, 62, 989–999, https://doi.org/10.1017/RDC.2020.18, 2020.

Sarnthein, M., Küssner, K., Grootes, P.M., Ausin, B., Eglinton, T., Muglia, J., Muscheler, R.,Schlolaut, G. Plateaus and jumps in the atmospheric radiocarbon record – Potential origin and value as global age markers for glacial-to-deglacial paleoceanography, a synthesis. Climate of the Past, 16, 2547–2571, doi: 10.5194/cp-16-2547-2020, 2020.

,,,,,,,,,,,,,,,,,,,,,,,,,,,,,,,,,,,,,,,,,,,,,,,,,,,,,,,,,,,,,,,,,,,,,,,,,,,,,,,,,,,,,,,,,,,,,

---

## Author Comment (AC3)

Dr. Michael Sarnthein, Prof. emer.                                    26 April 2022
Institut für Geowissenschaften
University of Kiel
Olshausenstr. 40
24098  K i e l,  Germanys
<michael.sarnthein@ifg.uni-kiel.de>

**COMMENTS to Review #2**

**by Michael Sarnthein and Pieter M. Grootes (authors)**

In this manuscript, the authors seek to establish the existence and timing of 'plateaus' in the atmospheric radiocarbon record, and to demonstrate that these are also present in marine records from around the world. On this basis, the authors seek to argue that radiocarbon plateaus identified in marine records can be stratigraphically aligned to correlative plateaus identified in the atmospheric record, allowing calendar ages to be transferred to the marine records (and therefore allowing for 'marine reservoir age' offsets to be determined). This method of chronostratigraphic alignment has been termed 'plateau tuning' (PT).

This paragraph nicely describes PT. This was the topic of Sarnthein et al., 2020, and of the objections of Bard and Heaton, 2021.

This manuscript is quite unusual, as it does not appear to advance any new observations/data, arguments, models or insights. Some adjustments are made (again) to the proposed timing of plateaus identified in the atmospheric radiocarbon record, but this does not really make any difference to what has been proposed by the authors in several papers since 2007.

**1. - Indeed the present manuscript does not provide new primary data. Different from the view of Rev.#2, however, our manuscript presents three major lines of "new evidence" that deserve publication, **(1)** a novel confirmation of the authenticity of atmospheric $^{14}$C plateau structures by means of a Bayesian spline plot in $\Delta^{14}$C/age space (courtesy of Bard & Heaton, 2021 (B&H), that now are clearly reproducing the structures of the Suigetsu atmospheric $^{14}$C record independently identified by our previous approaches, especially when all techniques use the updated Bronk Ramsey et al-2020 Suigetsu time scale. **(2)** We provide an adaption of the absolute age of all plateau boundaries in our marine $^{14}$C records, now solely based on a revised age control only published by Bronk Ramsey et al. (2020), that was coeval with the publication our CP synthesis article. The age revision has now been applied to all 19 ocean sediment records. This is crucial for the validity of PT, that is, for any proper use of plateau boundaries as global age tie points. (3) With great, yet unpublished detail our manuscript is meeting the unfounded allegation, also based on a misunderstanding of the importance of a suite of plateaus (pointed out in our comments but ignored), that sediment distortions by differential bioturbational mixing may form a major source of "fake" $^{14}$C plateaus.**

A recent 'review' of the 'PT method' and its results was published by the authors just last year in this same journal. Primarily, it seems, the manuscript seeks to publish a rebuttal of a prior piece of work produced by Bard & Heaton (B&H) that was also reviewed and published in Climate of the Past last year. The latter was also accompanied by several pages of commentary by Sarnthein and Grootes, which was in turn responded to by Bard and Heaton over the course of the discussion phase of the manuscript.

**2.- Since many arguments in our commentary were simply ignored by B&H (2021) a partial rebuttal of B&H theses was unavoidable in the present manuscript.**

Unfortunately, I find it impossible to recommend that this manuscript be accepted for publication. There are three main reasons for this: 1) it does not appear to present an original piece of research, and insofar as it presents adjustments, they are not important enough for publication on their own merit;

**3.- As said before, our manuscript shows that the centennial atmospheric $^{14}$C structures obtained using three different techniques are largely the same. The primary data may not be new but the outlined agreement, not pointed out by B&H, is telling a new perspective.**

2) its arguments against B&H are not coherent (regardless of whether or to B&H are correct);

**4.- The paper does not want to argue - again- against B&H. It just aims to show that the Suigetsu atmospheric $^{14}$C data set, though noisy and with limited coverage, can provide an authentic centennial-resolution signal of global significance extending beyond 14 ka, the present range of continuous tree-ring data. This data set can thus provide a valuable correlation target for the interpretation and global correlation of ocean sediment records.**

3) the vast bulk of figures and tables referred to in the manuscript are included in a 'supplement' that has not actually been produced/included. On the latter point, the promise of a compilation of all the available PT data in useful tables would have been at least one welcome contribution: but it turns out that the intention of the authors was to include ~20 disparate data tables that are already available on PANGAEA and that are not at all useful in reproducing the PT data that have been published to date by the authors (it took me days to do this, and the results are not the same as what the authors have published in many cases, which is both worrying and annoying).

**5.- All data tables necessarily contain the same primary depth and $^{14}$C information. However, a new Suigetsu time scale means new imported data and, potentially, a new correlation of plateaus. Those will be different and provided an upgrade replacing the earlier ones. Our recent compilation of ~20 data tables in a supplement has also been stored at PANGAEA under "https://doi.org/10.1594/PANGAEA.940604". Once this manuscript may be accepted, they are given with the explicit intention to replace (though properly cite) age tables previously published, somewhat diverging data tables that are obsolete after revision of the reference age scale and three minor revisions of plateau definition. All tables represent the same scheme of presentation. In addition, the tables of course need to take care of some local specialties of a sediment site recovered, such as listing paired benthic $^{14}$C ages in case available. Otherwise, we see no disparity. Minor differences in the sequence of $^{14}$C and sediment properties listed have now been adjusted.**

The fact that the PT data (and associated MRA etc.) that have been produced by the authors over several years, and presented in a series of 'global synopsis' papers, cannot be easily reproduced by others using the multitude of available data tables, is particularly worrisome.

We do not see a problem of a multitude of data tables, since the present set of tables will be clearly marked at PANGAEA as latest version 2022 and/or "latest state of the art" tables.

The same can be said for the fact that only one (?) PT study exists that does not include the authors of this study (the champions of the PT approach). Incidentally, this might already answer the question of whether or not it is a 'trend setting' tool.

**6.- Thanks for the kind remark. We feel worried by the traditional hesitation to accept a new higher $^{14}$C variability, combined with the strong warning written by B&H, that has discouraged use of PT in oceanography. Though it is certainly good to question the sometimes controversial new results and demand substantiation, the statement by B&H that PT should be verified by independent research (which is correct) is counteracted by their listing of objections that are partly based on misconceptions.**

With regard to the second point raised above, the authors state that they reject the arguments of B&H based on the basis of how plateaus are identified (i.e. as 'sequences', like a sort of Morse code), and on the basis that B&H use a 1998 box model to support their arguments. Regardless of the validity of B&H's remarks, I don't see how either of these points represent a coherent basis on which to reject a criticism of the PT method, where that criticism is founded in large part on the proposed difficulty of objectively identifying plateaus (let alone sequences of plateaus) in a noisy marine radiocarbon record whose offset from atmospheric radiocarbon varies over time, as well as the proposal that sedimentary processes (such as simple - and highly likely - sedimentation rate changes during periods such as Heinrich Stadial 1, or the Younger Dryas) can also produce 'plateaus' in the 14C age-depth domain, without these being causally linked to atmospheric radiocarbon variability.

**7.- The difficulty of unambiguously identifying and correlating plateaus is real but also not new. When the GISP2 and GRIP ice cores in Greenland provided in their $^{18}$O records highly detailed evidence for large and rapid climate variability paleoceanography followed with $^{18}$O signals in plankton in higher-resolution ocean cores, the number of peaks and valleys in both ice and ocean records was high and sediment dating not very detailed. Moreover, the ice core record has low accumulation and different thinning in cold phases, while sediments often have higher accumulation in cold phases and are subject to sedimentation and bioturbation problems as detailed by B&H. Yet, the patterns of D/O 14-13, 12-9, and 8-5 (Bond cycles) could be identified in various expressions in ocean sediment cores and provided a valuable link for improved dating and ocean-atmosphere correlation.**

The situation for PT is more difficult, because the atmospheric and oceanographic $^{14}$C signals are less clear than the $^{18}$O signals. Yet, the principle of correlating a full **suite** of $^{14}$C fluctuations/plateaus is similar and makes it possible to correlate the 'good' plateaus of a plateau sequence when one or more were destroyed/falsified by the mechanisms discussed by B&H. Again, PT is a 'tool' to explore whether more environmental information can be obtained from a sediment record. However, it does not provide a simple cookbook but rather shows a direction of additional analyses and comparisons needed to substantiate an initial plateau tuning with a consistent picture of the local oceanography and global climate recorded in the sediment core. If this succeeds valuable information has been gained.

In addition, the claim that a 1998 box-model is somehow incorrect because of its vintage seems to miss the point: the key purpose of deploying such a model is surely to illustrate in a very simple way how the phasing and amplitude-attenuation of an input signal will be altered (filtered) in the ocean, depending on the timescale on which the signal can be communicated to the ocean, and the frequency/duration of the signal variability. You can do this with a very complex biogeochemical coupled ocean-atmosphere numerical model if you like, but if it did not show a simple phase-attenuation relationship like the box model, it would mean that the complex model had a problem! In fact, by playing around with numerical model outputs it can be shown that they do show the same principles as a 2 box-model, and that should not be surprising, as it is an expression of a simple and fundamental physical principle: parts of the ocean that have small MRA offsets (such as the tropical ocean, MRA ~400 14Cyears) can respond quickly and can pick up shorter fluctuations from the atmosphere, whereas parts of the ocean that have large MRA offsets will take longer to pick up the atmospheric signal (since a larger MRA means that the isotopic exchange timescale for that water is longer) and will pick up a smoothed and lagged response. The limits of applicability of the PT method could readily be analysed and qualified in such a theoretical context, but the authors don't do this unfortunately.

**8.- On the basis of a coupled ocean GCM Lohmann et al. (2020) clearly show that $^{14}$C reservoir ages vary over small scale ocean regions, different from the assumptions of a simple box model simulation.**

Ultimately, the manuscript sets out to answer the question posed in the title: "is the 'plateau tuning' (PT) approach a misleading approach or a trend-setting tool"? I would note that, at worst, PT could be both misleading and trend-setting, and my major concern is that the authors clearly wish for it to be the latter, but have not really (either in the present manuscript, or over the course of several publications that appear to present the same datasets repeatedly) demonstrated that the PT approach is indeed viable, either in theory or in practice.

**9.- For a first time the present manuscript is documenting the authenticity of plateau structures, i.e., a major basis in support of the PT method. Admittedly, trend-setting' is may be a bit optimistic although new methods to analyze complex data often become quite trendy. 'Misleading' should not be possible if the researchers using PT 'do their homework' and carefully collect all circumstantial evidence they can to falsify interpretations till they are left with one that is verified by all available data.**

As suggested above, this is not to say that some sort of defence cannot be made, in theory at least. But the authors (still) have not managed to do this. My own view is that the chronostratigraphic principles that the authors wish to apply are not completely crazy: yes, the atmospheric radiocarbon record has 'wiggles' and these would be transferred to other reservoirs that exchange CO2 with the atmosphere rapidly enough to pick them up. However, the conditions under which these wiggles can be recorded in other reservoirs, such as the ocean, and the biases (in amplitude and phasing especially) that will inevitably and predictably arise (even prior to the complications of sedimentation changes, bioturbation, sampling/analytical noise etc.) need to be accepted and addressed by the authors at some point if this debate is to move in a useful direction.

**10.- Complications of regional ocean circulation changes like local upwelling, sediments rate changes, bioturbation, minimum sampling density and analytical noise have extensively been discussed and minimum qualities defined in our 2020 synthesis (and various papers since 2007), admittedly labor-intensive to read. A renewed lengthy repetition of all this reasoning appears unjustified and can now been avoided by citation of the synthesis paper.**
If there is a discrepancy between observations and theoretical predictions it can be that the interpretation of the observations is wrong (implied here). It is also possible that the system knowledge formalized in the model was still incomplete and that the level of detail used the model used, that was sufficient to answer research questions at the time it was developed, no longer can address the present problems. Considering the enormous gains in oceanographic knowledge over the

past decades and the work of e.g. Lohmann et al (2020), our guess is that a reevaluation of the theoretical restrictions and biases posed by local oceanography on local signals may be very valuable to move the debate.

**11.- Continuing our PT research, we now plan to add a statistical "BINNED correlation coefficient" to test the quality of correlation between the atmospheric reference record and each $^{14}$C record derived from ocean sediments as listed in supplement Tables S1 - S20.**

I can think of a variety of ways to test the PT method in theory (using models), and in practice using data, and I wonder why the authors have never done something similar.

**12.- By now we have been not able to generate a model to test the PT method in theory, since the variation of $^{14}$C reservoir ages follows a broad and highly complex multitude of factors of ocean circulation and carbon exchange, possibly a target for a future follow-up project of the group like that of Lohmann et al. (2020). Also, the reviewer ignores 15+ years of continued application and testing of PT in practice, which also served to gradually convince ourselves of its use. The testing in theory, using models, is easier said than done. First highly specialized modeling skills are needed and secondly detailed oceanographic insights are needed to improve models to the state that they can usefully provide local information over time interacting with climate. The authors would have loved to do this, but it is far beyond their reach.**

If a scientific study that achieved such goals was produced, it would be a welcome and useful addition to the literature (as B&H has proven to be, insofar as it stimulates critical thinking). Such a study would best come from the authors of the present study, who appear to be the main (if not the only?) champions of the PT method; however, this is not what the current manuscript provides.

For all arguments listed above, we like to plea that the present manuscript may be given a fair chance of publication in CP, certainly after a number of minor and major additions and revisions of the manuscript.

REFERENCES

Bard, E. and Heaton, T.J.: On the tuning of plateaus in atmospheric and oceanic 14C records to derive calendar chronologies of deep-sea cores and records of 14C marine reservoir age changes. Climate of the Past, 17, 1701–1725. https://doi.org/10.5194/cp-17-1701-2021, 2021.

Bronk Ramsey, C., Heaton, T.J., Schlolaut, G., Staff, R.A., Bryant, C.L., Lamb, H.F., Marshall, M.H., Nakagawa, T.: Reanalysis of the atmospheric radiocarbon calibration record from Lake Suigetsu, Japan. Radiocarbon, 62, 989–999, https://doi.org/10.1017/RDC.2020.18, 2020.

Lohmann, G., Butzin, M., Eissner, N., Shi, X., Stepanek, C.: Abrupt climate and weather changes across time scales. Paleoceanography and Paleoclimatology, 35, e2019PA003782, https://doi.org/10.1029/2019PA003782, 2020.

Sarnthein, M., Küssner, K., Grootes, P.M., Ausin, B., Eglinton, T., Muglia, J., Muscheler, R.,Schlolaut, G. Plateaus and jumps in the atmospheric radiocarbon record – Potential origin and value as global age markers for glacial-to-deglacial paleoceanography, a synthesis. Climate of the Past, 16, 2547–2571, doi: 10.5194/cp-16-2547-2020, 2020.

Sarnthein, Michael; Grootes, Pieter Meiert.: World Ocean $^{14}$C plateau tuning. PANGAEA, "https://doi.org/10.1594/PANGAEA.940604", 2022.

'''''''''''''''''''''''''''''''''''''''''''''''''''''''''''''''''''''''''''''''''''''''''''''''''''''''''''''''''''''''''''''''''''''''''''''''''''''''''''''''''''''''''''''